

# Revisiting temperature sensitivity: How does Antarctic precipitation change with temperature?

Lena Nicola[1,2,3], Dirk Notz[1,4], and Ricarda Winkelmann[2,3]

[1] Center for Earth System Research and Sustainability (CEN), Institute of Oceanography, Universität Hamburg, Germany
[2] Potsdam Institute for Climate Impact Research (PIK), Member of the Leibniz Association, P.O. Box 60 12 03, D-14412 Potsdam, Germany
[3] University of Potsdam, Institute of Physics and Astronomy, Karl-Liebknecht-Str. 24-25, 14476 Potsdam, Germany
[4] Max Planck Institute for Meteorology, Hamburg, Germany

**Correspondence:** Lena Nicola (lena.nicola@pik-potsdam.de), Ricarda Winkelmann (ricarda.winkelmann@pik-potsdam.de)

**Abstract.** With progressing global warming, snowfall in Antarctica is expected to increase, which could counteract or even temporarily overcompensate ice-sheet mass losses through increased ice discharge, calving and melting. For sea-level projections it is therefore vital to understand the processes determining snowfall changes in Antarctica. Here we revisit the relationship between Antarctic temperature changes and precipitation changes, identifying and explaining regional differences and deviations from the theoretical approach based on the Clausius-Clapeyron relationship. Analysing the latest estimates from global (CMIP6) and regional (RACMO2.3) model projections, we find an average increase of 5.5 % in annual precipitation over Antarctica per degree of warming, with a minimum sensitivity of 2 % K$^{-1}$ near Siple Coast, and a maximum sensitivity > 10 % K$^{-1}$ at the East Antarctic Plateau region. This large range can be explained by the prevailing climatic conditions, with local temperatures determining the Clausius-Clapeyron sensitivity that is counteracted in some regions by the prevalence of the coastal wind regime. We compare different approaches of deriving the sensitivity factor, which in some cases can lead to sensitivity changes of up to 7 % for the same model. Importantly, local sensitivity-factors are found to be strongly dependent on the warming level, suggesting that some ice-sheet models which base their precipitation estimates on parameterizations derived from these sensitivity factors might overestimate warming-induced snowfall changes, particularly in high-emission scenarios. This would have consequences for Antarctic sea-level projections for this century and beyond.

## 1 Introduction

Over the past decades, the Antarctic Ice Sheet has been losing mass at an accelerating pace (IMBIE Team, 2018; Rignot et al., 2019) and is increasingly contributing to sea-level rise (Fox-Kemper et al., 2021). Melting ice from the Antarctic Ice Sheet has risen global sea levels by 7.4±1.5 mm between 1992 and 2020, caused by the total ice loss of 2671±530 Gt over that period (Otosaka et al., 2022). Due to on-going melt, global sea levels are committed to rise for centuries to come (Levermann et al., 2013; Golledge et al., 2015).

The Antarctic Ice Sheet could however not only be a contributor to sea-level rise but may even slow down the rise in sea level by storing additional mass through increased snowfall (Seroussi et al., 2020). Antarctic precipitation is by far the most





important positive contributor to the overall mass balance of the Antarctic Ice Sheet. The balance between snow accumulation in the interior minus the surface ablation (wind transport, sublimation, very low surface melt) and the ice loss through calving
and sub-shelf melting determines the magnitude and pace of the Antarctic contribution to past and future global sea-level rise.

The uncertainty of the Antarctic sea-level contribution in modelling studies generally arises both from the uncertainty in the external (climate) forcing as well as from uncertainties in representing the governing processes and their relevant parameters in models (e.g., Rodehacke et al., 2020; Seroussi et al., 2020). Parts of this uncertainty arise from our limited understanding how Antarctic precipitation is changing with warming and how the change in snowfall rates can be incorporated into ice-sheet
models. Addressing this uncertainty is the focus of this contribution.

Present-day observations of Antarctic precipitation are sparse and regional climate models disagree strongly in their estimates of annual surface mass balance (Mottram et al., 2021). For what is known, Antarctica is as dry as desert climates (annual precipitation < 250 mm, Sikka, 1997) and is therefore often referred to as a *Polar desert*. Palerme et al. (2014) obtained continent-wide snowfall rates through satellite-based radar and estimated a mean annual snowfall of 171 mm from August
2005 to April 2011. Roussel et al. (2020) state an annual snowfall of roughly 186 mm w.e. per year.

Most of the ice mass lies in the interior, but precipitation in Antarctica is concentrated at the ice-sheet margins. Annual precipitation is exceeding $1000 \, \mathrm{mm \, yr^{-1}}$ in coastal parts of West Antarctica, near Wilkes Land as well as at the Antarctic Peninsula (see Panel (a), left, Fig.1). In the interior of the ice sheet, mean annual precipitation is below $50 \, \mathrm{mm \, yr^{-1}}$.

Despite the little annual snowfall, mass gains through snowfall have exceeded mass losses from the Antarctic Ice Sheet
between 2003-2008 (Zwally et al., 2015). Model simulations show that Antarctic snowfall may increase significantly in a warming climate, and could thus partly buffer the warming-induced ice loss (Bracegirdle et al., 2008; Frieler et al., 2015; Rodehacke et al., 2020). While insignificant changes of snowfall were reported from 1957 to 2006 (Monaghan et al., 2006), Medley and Thomas (2019) find that snow accumulation has been increasing by 1.1 mm per decade between 1901 and 2000 and by 2.5 mm per decade after 1979, mitigating sea-level rise by about 10 mm since 1901.

It is hypothesised that Antarctic snowfall increases with temperature according to the Clausius-Clapeyron relationship (Clapeyron, 1834; Clausius, 1850), describing the saturation water vapour pressure, $e_s$, as a function of temperature, $T$. This hypothesis is based on the assumption that Antarctic precipitation is solely driven by temperature and the associated availability of moisture in the atmosphere. It is assumed that Antarctic snowfall therefore increases with the same sensitivity as the general capacity of the air to hold moisture, which is given by the saturation water vapour pressure $e_s$ beyond which water vapour con-
denses and can thus potentially precipitate as snow in Antarctica. Held and Soden (2006) introduce the Clausius-Clapeyron relationship as

$$\frac{d\ln e_s}{dT} = \frac{L}{R_v T^2} \equiv \alpha(T) \tag{1}$$

with $L$ being the latent heat of vaporization and $R_v$ the specific gas constant for water vapour. $\alpha(T)$ in Equation (1) is the sensitivity parameter, translating the change in temperature into a relative change in saturation water vapour pressure. With $L =$
$2.5 \times 10^6 \, \mathrm{J \, kg^{-1}}$, $R_v = 461 \, \mathrm{J \, K^{-1} \, kg^{-1}}$ and a mean temperature of the lower troposphere of $T \sim 260 \, \mathrm{K}$, a global approximation yields $\alpha \approx 7 \, \% \, \mathrm{K^{-1}}$. That means that atmospheric moisture content generally rises by 7 % per 1 K of warming (Hartmann,



2016). Using a continent-wide mean annual air temperature of T = 239.55 K (1981-2000 mean of ERA5-Land reanalysis data), the sensitivity factor, $\alpha(T)$ can be approximated as 9.45 % K$^{-1}$ for Antarctic conditions.

Projections of regional climate models show a wide range of snowfall changes in the coming decades depending on the model input (Kittel et al., 2021). Simulations of the regional model RACMO2.3, which is often used as input of numerical ice-sheet models (e.g. in Garbe et al., 2020; Seroussi et al., 2020), project that mean annual Antarctic precipitation will increase from approximately 189 mm w.e. yr$^{-1}$ in 1981-2000 to 289 mm w.e. yr$^{-1}$ at the end of the century for the SSP5-85 scenario. This corresponds to an increase by $+52.43\%$ for the simulated mean temperature increase of 6.7 K. In these simulations, precipitation increases most in coastal areas, but also rises in the interior (Panel (a), right, Fig. 1).

Global coupled climate models participating in the Coupled Model Intercomparison Project Phase 6 (CMIP6) show differently strong responses of Antarctic precipitation to temperature changes in the 21st-century. End-of-century (2081-2100) Antarctic surface-air temperatures are projected to change relative to present-day conditions (1981-2000) by $1.6 \pm 0.8$ K, $2.7 \pm 0.9$ K and $4.7 \pm 1.4$ K, for a low (SSP1-26), intermediate (SSP2-45) and high emission scenario (SSP5-85). For these temperature changes, annual precipitation is projected to increase by $9.7 \pm 7.3$ %, $15.8 \pm 8.1$ %, and $28.8 \pm 12.6$ %, respectively.

Because it is numerically expensive and technically challenging to couple global atmosphere-ocean general circulation models to an interactive ice sheet, standalone ice-sheet models are usually used that often employ a scaling approach to translate changes in air temperature to changes in Antarctic precipitation. In ice-sheet models, precipitation can be scaled with temperature or temperature anomalies, using sensitivity factors (%K$^{-1}$) given by the existing literature. This approach is often used in long-term projections, where regional climate model estimates are not available: Albrecht et al. (2020) for instance

used different values for the sensitivity factor to perform glacial-cycle simulations and to test for parameter sensitivity. Quiquet et al. (2018) scale the surface mass balance with a sensitivity factor, assessing Antarctic Ice Sheet changes for the last 400 kyr. Huybrechts (2002) deduce the precipitation and basal melt rate from simple temperature relationships for performing glacial cycle simulations. Rodehacke et al. (2020) scale precipitation with temperatures, estimating Antarctica's sea-level contribution when using different precipitation parameterizations such as CMIP5 model output or constant scaling factors inside the ice-

sheet model.

     Generally, snowfall in Antarctica depends on a complex interplay of processes. Not only moisture availability and temperature play a crucial role, but also local wind regimes (Grazioli et al., 2017), the occurrence of atmospheric rivers, and large scale atmospheric variability (Nicolas et al., 2017; Wille et al., 2019; Maclennan et al., 2022). Synoptic scale features, such as cyclones and fronts, generally influence coastal precipitation (Bromwich, 1988). The long-term evolution of precipitation is

found to be dominated however by thermodynamic changes (Uotila et al., 2007; Krinner et al., 2014; Seneviratne et al., 2021).

     It is known that the Antarctic Ice Sheet may gain mass under warming due to increased snowfall. Such increase is expected to generally follow a given rise in temperature according to the Clausius-Clapeyron relationship. Already Robin (1977) has proposed a linear relationship between water vapour pressure over ice and temperature, whilst concluding that this is "an empirical approximation to observations, rather than a natural law". Krinner et al. (2007), Bengtsson et al. (2011), Ligtenberg

et al. (2013) or Agosta et al. (2013) use changes in surface mass balance (SMB) to estimate sensitivity factors between precipitation and temperature, while Frieler et al. (2015), Fudge et al. (2016) and Medley and Thomas (2019) use changes in

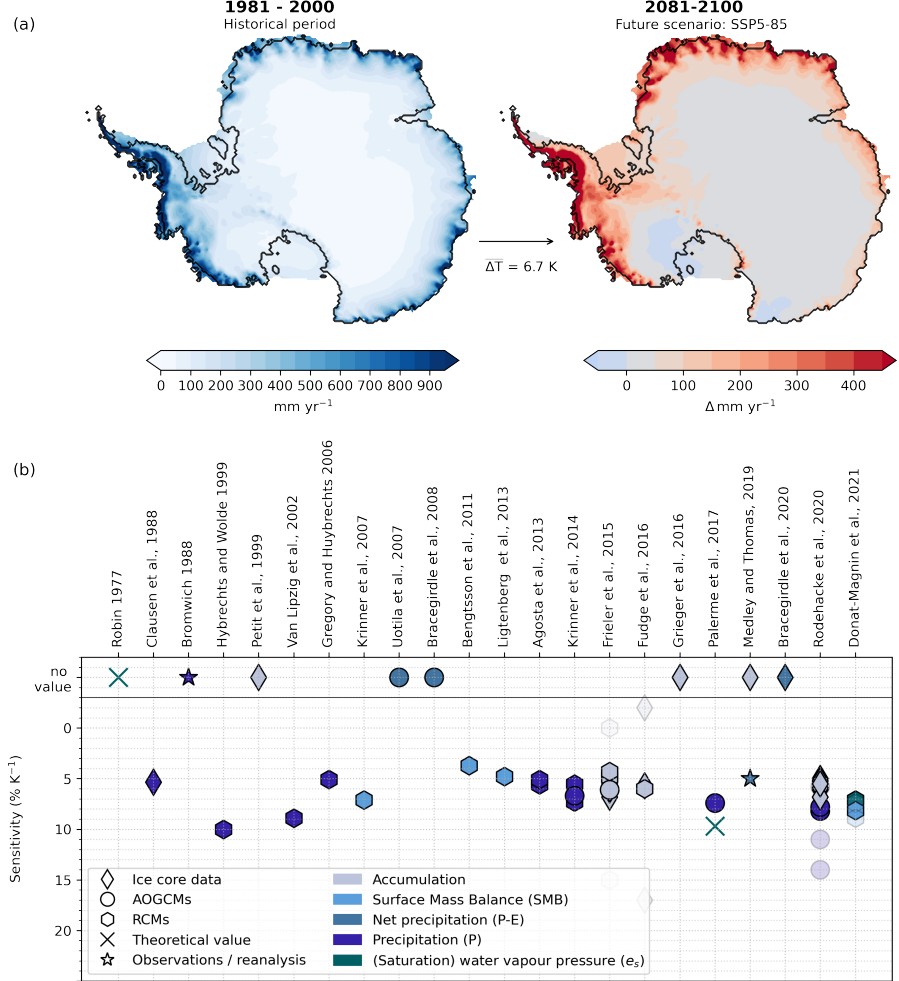

**Figure 1. How is Antarctic precipitation changing with warming? (a)**: Change in mean annual precipitation as simulated with the regional climate model RACMO2.3 (left) for the historical period (1981-2000) and (right) projected for the end of this century (2081-2100) under the SSP5-85 scenario. On average, Antarctic-wide temperatures change by 6.7 K between the two time periods. **(b)**: Literature values from ice-core data (diamonds), AOGCMs (circles), RCMs (hexagons) as well as observations/reanalysis (stars), for the sensitivity of precipitation, net precipitation, accumulation and surface mass balance (often linked to the sensitivity of saturation water vapour pressure) to warming (given in $\% \, K^{-1}$). Upper row shows studies assessing the relationship but without quantifying such sensitivity factor. Translucent markers indicate extreme values found for example within ice cores (Fudge et al., 2016) or in modelling results (Frieler et al., 2015; Rodehacke et al., 2020; Donat-Magnin et al., 2021).

snow accumulation to derive relative changes in snowfall per degree of warming ($\% \, K^{-1}$). Other studies have determined a sensitivity of net precipitation to warming, meaning precipitation minus evaporation (commonly denoted P-E), to also account for an increase in evaporation rates (Uotila et al., 2007; Bracegirdle et al., 2008). Palerme et al. (2017) use changes in total





precipitation (P) estimates, focusing on the increase of snowfall (+ rain) with warming. Several more studies have analysed a
potential connection of mass gains and atmospheric warming, see Fig. 1, but have not estimated a sensitivity factor in the form
that is discussed here ($\%\,\mathrm{K}^{-1}$). As data sources for estimating the sensitivity of precipitation, existing studies have incorpo-
rated ice core data (Petit et al., 1999; Frieler et al., 2015; Fudge et al., 2016), ice core data combined with reanalysis (Medley
and Thomas, 2019), AOGCM output partaking in early CMIP initiatives (Uotila et al., 2007; Bracegirdle et al., 2008), CMIP3

(Gregory and Huybrechts, 2006; Krinner et al., 2014) and CMIP5 (Frieler et al., 2015; Grieger et al., 2016; Palerme et al.,
2017; Rodehacke et al., 2020), or high-resolution, regional or paleoclimate model output (Krinner et al., 2007; Agosta et al.,
2013; Ligtenberg et al., 2013; Krinner et al., 2014; Frieler et al., 2015; Donat-Magnin et al., 2021).

Overall, the sensitivity factors assessed from the literature vary roughly between 4 and 10 $\%\,\mathrm{K}^{-1}$, see Fig. 1, with extreme
values for the change in snow accumulation found in parts of ice cores (Fudge et al., 2016) and certain modelling studies

(Frieler et al., 2015; Donat-Magnin et al., 2021).

In this paper, we update previous continent-wide estimates of the sensitivity factors of Antarctic precipitation to temperature
based on the latest available model data and reconcile it with previous approaches (Section 3). We show how and explain why
these sensitivity factors differ strongly across the ice sheet (Section 4). We conclude that the scaling approach often used in
ice-sheet models should be revised, depending on the chosen application (Section 5).

## 2    Methods


In our study we revisit the temperature-dependency of snowfall changes on the Antarctic Ice Sheet. We use a linear least-
squares regression analysis to determine the sensitivity factor $\alpha$ that describes how Antarctic precipitation is changing with
temperature. This approach follows the general definition of the Clausius-Clapeyron relationship (Eq. 1). Sensitivity factors
have been commonly estimated using relative changes of precipitation or accumulation (changes in % compared to a reference

period) and values of warming ($\Delta$T, e.g. Frieler et al., 2015; Fudge et al., 2016; Palerme et al., 2017). For example, Frieler et al.
(2015) use changes in warming and relative changes in precipitation compared to 1850-1900. Krinner et al. (2007), Krinner
et al. (2014) and Palerme et al. (2017) compare changes between two states, e.g. the end of the 20th-century versus the end of
the 21st-century.

In our analysis, we follow the Clausius-Clapeyron theory (Eq. 1) more closely, applying the regression analysis to log-

scaled mean annual precipitation and the annual temperature time series. This makes our approach independent of the chosen
reference period. (Donat-Magnin et al., 2021) use a similar approach, but they do not account for the full length of the available
timeseries.

We perform a sensitivity analysis in three ways (see also Figure A1 for a graphical representation):

1. Continent-wide regression - We first average over available temperature and precipitation fields across the entire Antarc-

125       tic continent and then obtain a sensitivity value from the least-squares linear regression of the time series of continent-
wide annual temperature and log-scaled precipitation.





2. Grid-point regression - We first perform the least-squares linear regression with the local time series of annual temperature and log-scaled precipitation for every grid point. In this regression the predictor arrays are the time series of temperature for each grid point respectively. This yields a spatial distribution of scaling factors. For comparing these estimates to the continent-wide regression, these grid values are averaged over the ice sheet.

3. Spatial regression - For each time slice of the available data (x,y,time), we perform a linear regression with the data points from the spatial distribution of temperature and log-scaled precipitation (x,y). Here the predictor values are the 1440X1080 (lonxlat) grid points of temperature values for each time slice. The regression with the 1440x1080 grid points of precipitation then yields a new estimate of the sensitivity of how Antarctic precipitation follows local temperatures across the ice sheet. For analysing the change in this sensitivity with temperature, we perform a second linear regression with the mean annual temperature time series. This second step makes this approach distinct from the grid-point regression where only one regression is performed.

Our analysis is based on different types of data to robustly delineate the sensitivity of Antarctic precipitation to temperature under present-day conditions, as well as their potential changes in the future.

While direct measurements are scarce and observational products such as the CloudSat data lack the needed resolution (Palerme et al., 2014), we use the ECMWF ERA5-Land reanalysis data (Muñoz Sabater, 2019) as a best estimate of present-day conditions in Antarctica. These reanalysis data provide spatially and temporally complete coverage of the historical and present-day evolution of precipitation and temperature patterns for Antarctica. The ERA5-Land reanalysis is provided through the Copernicus Climate Change Service (C3S) at the ClimateData Store and is available at a resolution of $0.1\,°\,\text{x}\,0.1\,°$ on a lon-lat grid at hourly resolution. We here use monthly averaged variables.

In addition, we analyse CMIP6 model data which is available from Earth System Grid Federation (ESGF, for example at https://esgf-data.dkrz.de). Where available, we use three shared-socioeconomic scenarios for characterising future climatic conditions in Antarctica: SSP1-26 as a low emission, SSP2-45 as intermediate and SSP5-85 as a high emission scenario (Riahi et al., 2017). We combine historical data that covers the period of 1850-2014 with projections for the years 2015 to 2100. A selection of models provide projections until the year 2300, including for the SSP1-26 scenario models CanESM5, IPSL-CM6A-LR, MRI-ESM2-0, and UKESM1-0-LL, and additionally for the SSP5-85 scenario models ACCESS-CM2 and MIROC-ES2L. We use the first available ensemble member of each CMIP6 model for analysis i.e. r1i1f1p1 in most cases. The nominal resolution of the CMIP6 ensemble differs substantially and lies between 50 km (CNRM-CM6-1-HR and GFDL-CM4) and 500 km (CanESM5). If possible we use the native model mask (through variable *sftlf*) to extract data for the Antarctic continent. For analysing the regional CMIP6 model mean of sensitivity factors, we regrid all models to a common 1440x1080 grid, following the highest resolution of the available models (GFDL CM4). We incorporate more than 30 different models for the analysis until 2100.

Adding to that, we use mean monthly values of near-surface temperature and precipitation data from the regional model RACMO2.3 for the years 1950 to the end of the 21st-century (van Meijgaard et al., 2008; Van Wessem et al., 2018). For the





future period 2015-2100, RACMO2.3 is here forced with CESM2 model output for the SSP5-85 scenario. The data is available at a 27 km resolution.

We analyse the full time series of yearly mean temperature and precipitation until the year 2100 (and in some cases until 2300). In order to obtain a 20th-century and a 21st-century reference period, we average values over the years 1981-2000 and 2081-2100, respectively. We use a twenty year average to reduce the impact of internal variability, following the approach in the recent IPCC AR6 WG1 report (Masson-Delmotte et al., 2021). Mean values of both time windows will be used to assess sensitivity factors as in Palerme et al. (2017).

## 3 Continent-wide scaling factors from regional and global climate model data

Analysing Antarctic temperature and precipitation from all available CMIP6 models over the time period 1850-2100, we find a sensitivity of Antarctic precipitation to temperature of approx. $\alpha = 5.5\,\%\,\mathrm{K}^{-1}$, which is independent of the chosen climate-change scenario and close to previous estimates. The statistical means across all individual CMIP6 sensitivities from the continent-wide regression are $5.48 \pm 1.17\,\%\,\mathrm{K}^{-1}$, $5.46 \pm 1.05\,\%\,\mathrm{K}^{-1}$ and $5.46 \pm 0.85\,\%\,\mathrm{K}^{-1}$ using the historical period and the three SSP-scenarios respectively (SSP1-26, SSP2-45, SSP5-85). Using the inter-model mean of Antarctic precipitation and temperature results in slightly higher values of $\alpha = 6.23\,\%\,\mathrm{K}^{-1}$, $5.94\,\%\,\mathrm{K}^{-1}$ and $5.71\,\%\,\mathrm{K}^{-1}$, see Fig. 2. RACMO2.3 data give a sensitivity factor of $\alpha = 6.37\,\%\,\mathrm{K}^{-1}$ using the SSP5-85 scenario with the available historical period from 1950 to 2100. $6.37\,\%\,\mathrm{K}^{-1}$ lies close to the upper end of the inter-model spread of the CMIP6 ensemble for the SSP5-85 scenario ($5.46 \pm 0.85\,\%\,\mathrm{K}^{-1}$) and the deviation could thus generally be explained by differing model characteristics.

The $R^2$-values of the performed linear regressions with the CMIP6 model data are generally highest for the SPP5-85 scenario with $R^2$ up to 0.94 for models CESM2-WACCM, CNRM-CM6-1-HR and CanESM5, see Fig. (2) for details. Note here that at the time of our analyses not all CMIP6 models provided all future scenarios. All model-specific scaling factors are summarised in Table A1. For the RACMO2.3 data we obtain a $R^2$ value of 0.92.

Our obtained sensitivity factor of approx. $5.5\,\%\,\mathrm{K}^{-1}$ is slightly lower than the CMIP5 estimate of $6.1\,\%\,\mathrm{K}^{-1}$ derived in Frieler et al. (2015). This can in parts result from differences in the CMIP6 versus the CMIP5 ensemble (Zelinka et al., 2020; Payne et al., 2021). Moreover, as described above we are using a log-based approach here rather than relative anomalies which also leads to slightly different estimates. Using the same approach as in Frieler et al. (2015) (i.e., anomalies wrt. 1890–1980), we obtain a mean sensitivity of $5.86 \pm 1.29\,\%\,\mathrm{K}^{-1}$. Individual model results from that analysis can be seen in Fig. A3. Quantifying the changes between the two reference periods, i.e., the end of the 20th vs. the end of the 21st-century, results in a higher sensitivity of $7.3\,\%\,\mathrm{K}^{-1}$, see Fig. A4, which is closer to the CMIP5 estimate in Palerme et al. (2017). This shows that the calculated sensitivity depends on the chosen analysis method.

For the extended CMIP6 projections until 2300, we use the model output by CanESM5, IPSL-CM6A-LR, MRI-ESM2-0, UKESM1-0-LL, ACCESS-CM2 and MIROC-ES2L. We find that the approach of estimating sensitivity factors from changes relative to a reference period shows a stronger bias for the SSP5-85 scenario, see Fig. 3. The sensitivity factors for e.g. the CanESM5 model results in $13.31\,\%\,\mathrm{K}^{-1}$ for the relative anomaly approach compared to $6.32\,\%\,\mathrm{K}^{-1}$ in our logarithmic ap-



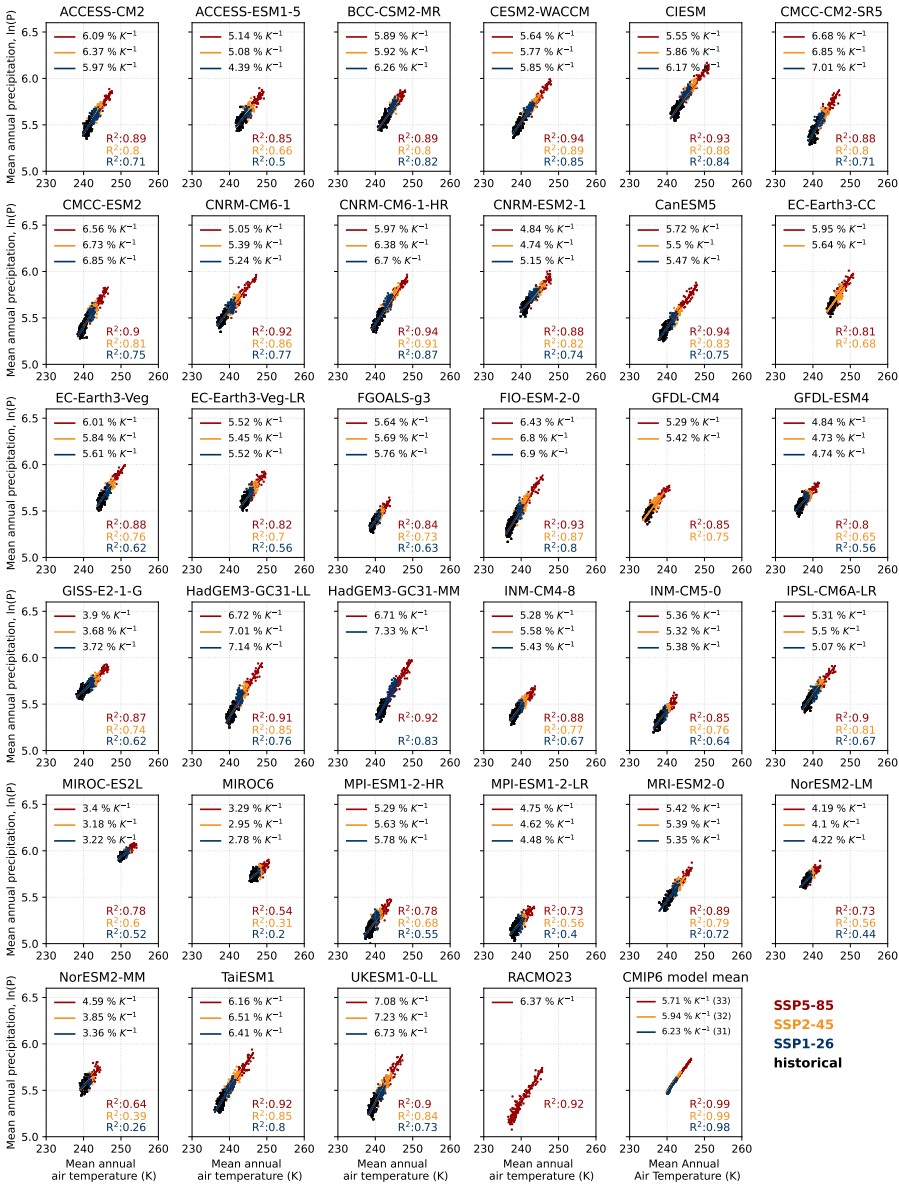

**Figure 2. Update on continent-wide scaling factors based on CMIP6 and RACMO2.3 21st-century projections.** Sensitivity factors are estimated over the period 1850-2100 for the CMIP6 ensemble by combining the historical period with three available SSP-scenarios (SSP1-26, SSP2-45 and SSP5-85), and over the period 1950-2100 for RACMO2.3. For the CMIP6 model mean, the numbers in brackets refer to the number of models incorporated into the analysis.

proach. The difference is due to the nature of the regression itself. In the relative anomaly approach we approximate the exponential function with percentage changes, which only holds for very small changes in the predictor variable, here increments of



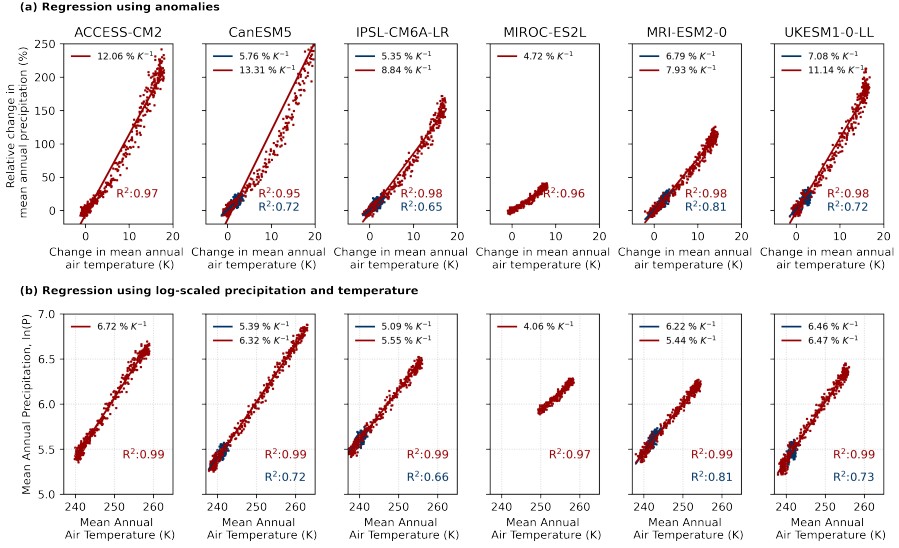

**Figure 3. Continent-wide scaling factors for CMIP6 models simulating Antarctic climate change until 2300.** Two different approaches for determining the sensitivity factor are used: **(a)** shows the results when estimating the sensitivity of Antarctic precipitation to temperature from anomalies wrt. to 1890-1980 as done in Frieler et al. (2015). **(b)** shows the results when using log-scaled precipitation with absolute temperature estimates in the regression analysis.

warming ($\Delta T$). If the chosen model shows strong warming, hence large $\Delta T$, the regression becomes inaccurate. Using output from climate models that show strong warming rates, i.e. that have a high equilibrium climate sensitivity, such as CanESM5 (Meehl et al., 2020), the relative anomaly approach thus significantly alters the results from the multi-model analysis. Our logarithmic approach on the other hand incorporates the exponential function directly in the regression analysis; this avoids a potential bias towards the models with higher climate sensitivity (sometimes referred to as the "hot model problem",see e.g.

(Hausfather et al., 2022).

## 4   Regional sensitivity factors differ across the ice sheet

Performing the grid-point regression of available CMIP6 model data shows that, across the ice sheet, sensitivity factors in certain regions are substantially different from the continental scaling factor of approximately 5.5 % K$^{-1}$ obtained in the previous section, see Fig. 4. This largely confirms findings by Rodehacke et al. (2020), showing that regional sensitivities can

differ substantially from the continent-wide scaling also in CMIP5. Note that we here use the time period of 1950-2100 to compare the spatial sensitivities with the results from RACMO2.3 (which are available for the same time period). Using the period from 1850 to 2100 only causes very minor differences in our results.

The spatial sensitivities obtained from the multi-model mean of the regridded temperature and precipitation fields show similar patterns across the different SSP-scenarios: In the Ross ice-shelf region there are very low sensitivity factors while



in the interior, factors go up to more than 10 % K$^{-1}$. Sensitivity factors are on average around 2% higher in East Antarctica
than in West Antarctica (here given roughly by the 40° W / 320° E and 180° W / E longitudes as lateral boundaries). We
here acknowledge more sophisticated ice dynamical definitions i.e. that are derived from ice divides of individual ice drainage
basins (Rignot et al., 2011; Zwally et al., 2012)). For the three chosen SSP-scenarios, the mean area-weighted factors are 7.69,
7.38, and 7.15 % K$^{-1}$ for East Antarctica and 5.68, 5.36 and 5.26 % K$^{-1}$ for the West Antarctic Ice Sheet, respectively. The

mean R$^2$-values for the EAIS are higher than for the WAIS (R$^2$ = 0.79, 0.86, 0.94 vs. R$^2$ = 0.64, 0.72, 0.85 for the respective
future scenarios, compare Fig. A5). The difference between the two ice sheets could results from the low sensitivity factors
found near Siple Coast, where the linear regression performs very poorly and skews the mean for the WAIS to lower values.
This could be for instance due to a prominent area of converging katabatic winds (Parish and Bromwich, 2007), that could
diminish precipitation at the coast (Grazioli et al., 2017).

Comparing sensitivity factors across the ice sheet with the respective present-day temperatures (see Fig.A2), allows us to
explain much of the spatial patterns: Higher sensitivity factors are generally found in regions with lower temperatures. This is
consistent with the theory, as the Clausius-Clapeyron relationship in Eq. (1) gives higher values of $\alpha$ for colder temperatures.
Local temperatures especially in East Antarctica can reach well below the mean annual air temperature of 239.55 K / -33.6 °C
(1980-2000 mean from ERA5-Land reanalysis), which would result in a theoretical sensitivity factor of 9.45 % K$^{-1}$, which lies

close to the derived values in model data analysis. The relationship between local temperatures and sensitivity factors is most
pronounced on the East Antarctic plateau where the influence of coastal winds is considered to be less significant (Bromwich,
1988).

While the overall spatial pattern is robust for the different climate change scenarios, the sensitivity factors are generally
lower for the high-emission SSP5-85 scenario, and higher for the low-emission SSP1-26 scenario. This tendency can be seen

in the local factors across the ice sheet, with the difference between scenarios being particularly pronounced in East Antarctica.
The tendency of lower sensitivity factors for higher emissions is even more apparent when averaging over the scaling factors
for each scenario (see Panel (c) in Fig. 4): we find a mean area-weighted scaling factor of 7.19, 6.86 and 6.67 % K$^{-1}$ for
SSP1-26, SSP2-45 and SSP5-85 scenario, respectively. This is consistent with the RACMO2.3 mean scaling factor of 6.61 %
K$^{-1}$ for the SSP5-85 scenario.

This mean of the spatially resolved sensitivity factors of the CMIP6 model data is thus higher than the continent-wide
estimate of 5.5 % K$^{-1}$, which was independent of the chosen warming scenario. (Note the difference between the mean area-
weighted scaling factor and the continent-wide scaling factor, see methods section). This is likely due to many local factors
being averaged out when generating the continent-wide temperature and annual precipitation time series. For the RACMO2.3
model we find small regions of high sensitivity factors in parts of Dronning Maud Land and around the Filchner-Ronne Ice

Shelf that are not visible in the CMIP6 model results, see Fig. 4. We believe this is due to local dynamic effects which are
incorporated in the regional climate models and are not resolved in the CMIP6 models. This is consisted with regional studies,
finding regional scaling factors of 7.4 to 8.9 % K$^{-1}$ for the Amundsen Sea region (Donat-Magnin et al., 2021).

We find an even stronger difference between a 'cold' and a 'warm' future scenario when examining the local sensitivity
factors from those models that simulate Antarctic precipitation and temperature until 2300 (see Fig. 5). The results of the low-



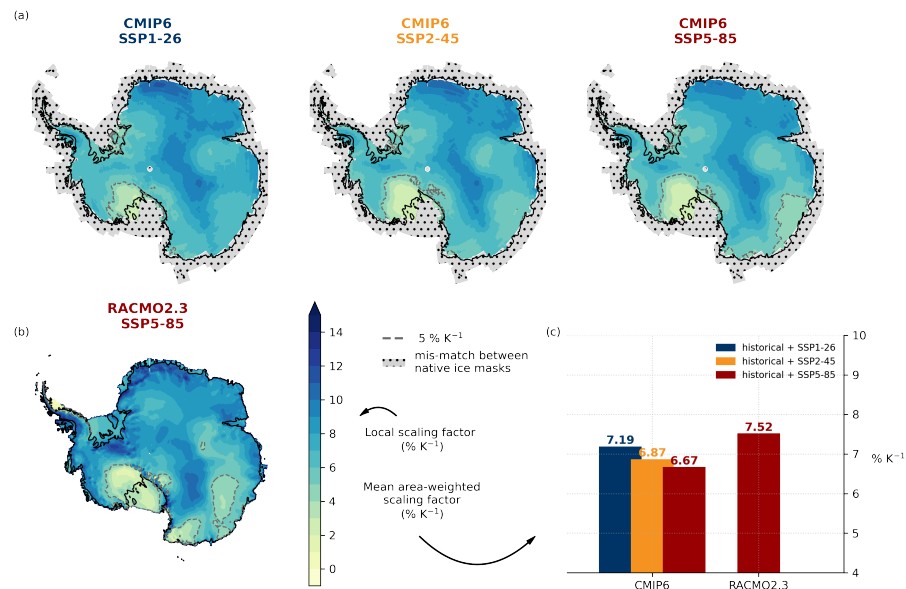

**Figure 4. Differences in sensitivity factors across the ice sheet derived from 21st century projections. (a)** Spatially-resolved sensitivities for the CMIP6 model mean for each SSP scenario. The point-wise regression is based on the period 1950 to 2100 by combining the historical period with the SSP1-26, SSP2-45 and SSP5-85 future scenarios, respectively. **(b)** Spatially-resolved sensitivities for RACMO2.3 model data, which was forced by CESM2 with SSP5-85 forcing (1950-2100). Dashed lines in maps indicate the 5 % K$^{-1}$ contour line, which refers to a commonly used sensitivity factor in ice-sheet modelling, such as in Garbe et al. (2020). Hatched regions shows a mis-match between native ice masks of the CMIP6 ensemble which is excluded from the analysis. **(c)** Comparison of area-weighted mean sensitivities, averaged over the same area of interpolated CMIP6 and RACMO2.3 data.

(SSP1-26) and high emission scenario (SSP5-85) show a strong difference in temperature sensitivities across the ice sheet. For the SSP1-26 scenario, the area-weighted mean scaling factors across the ice sheet are > 8 % K$^{-1}$. For the warmer SSP5-85 scenario, we find much lower sensitivities. The differences in the area-weighted mean scaling factors between the two scenarios lie between 0.9 % K$^{-1}$ for CanESM5 and 2.9 % K$^{-1}$ for the MRI-ESM2-0 model. Here the CanESM5 model shows local warming of > 30 K by 2300 compared to present day, which leads to a strong reduction in sensitivity as expected from the definition of $\alpha$.

Our results highlight that when simulating changes in Antarctic mass balance in the future, we need to consider these local sensitivities of precipitation change to warming. Using spatially resolved scaling factors that depict the local conditions could improve projections of the Antarctic sea-level contribution.

As we find that local sensitivity-factors depend on the warming level, ice-sheet models which base their precipitation projections on parameterizations derived from these sensitivity factors might overestimate warming-induced snowfall changes, particularly in high-emission scenarios.





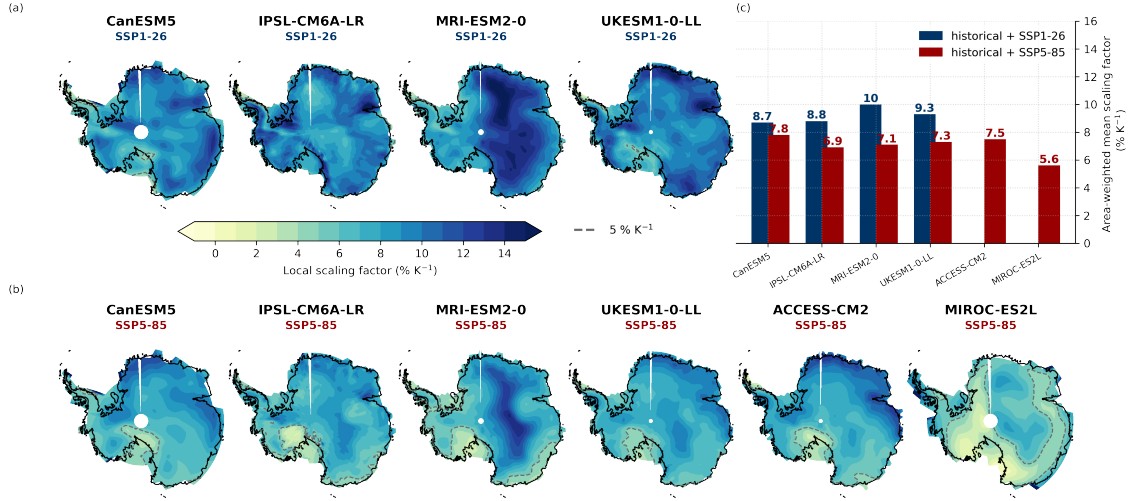

**Figure 5. Differences in sensitivity factors across the ice sheet derived from projections until 2300.** Results are given for individual CMIP6 models that were run until the year 2300 for **(a)** SSP1-26, and **(b)** SSP5-85. Dashed lines in maps indicate the 5 % K$^{-1}$ contour line, which refers to a commonly used sensitivity factor in ice-sheet modelling, such as in Garbe et al. (2020). **(c)** Comparison of area-weighted mean sensitivities.

## 5 Decrease in sensitivity with future warming

We further analyse to which degree the regional temperature distribution can explain the regional distribution of precipitation rates across the ice sheet. Using the present-day distribution of temperature and precipitation (based on the 1981-2000 mean from the ERA5-Land reanalysis, see Fig. 6 a), we find that the temperature pattern in Antarctica can explain roughly 75 % of the annual precipitation when assuming a linear relationship between the temperature and precipitation fields. The analysed sensitivity would result in a precipitation increase of 7.89 % per 1 K temperature difference across the ice sheet. The difference between the local precipitation rate estimated from the simple linear temperature relationship and the reanalysed precipitation is particularly low in the East Antarctic plateau above 3000 m altitude, see Fig. 6 (b).

We find that also this sensitivity factor changes over time: When repeating the analysis for each year from 1850 to the end of the 21st century (23rd century, where data is available), we find that the 20-year running mean sensitivity declines by -0.064 ($\pm$ 0.045), -0.060 ($\pm$ 0.036) or -0.065 ($\pm$ 0.039) points per degree of temperature rise in the SSP1-26, SSP2-45 and SSP5-85 scenario respectively (see Fig. 7). This decrease in sensitivity over time is especially strong for the simulations extending until year 2300.



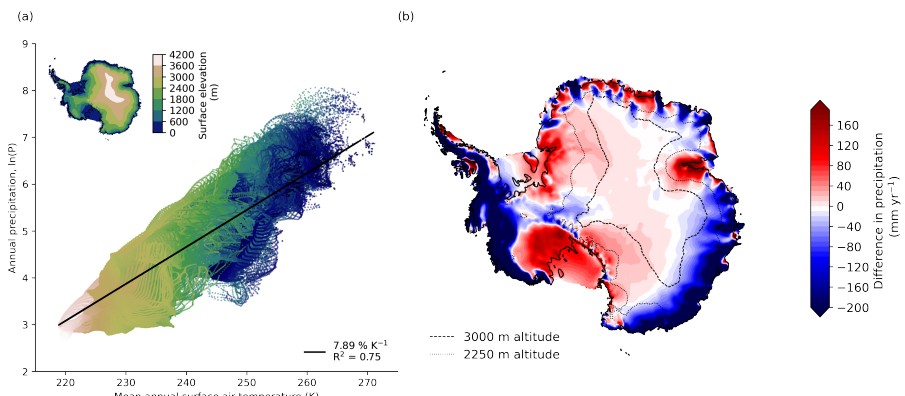

**Figure 6. Antarctic precipitation determined by local air temperatures across the ice sheet. (a)** Estimates of log-scaled precipitation against mean annual surface air temperatures for each grid point in the ERA5-land reanalysis (1981-2000 mean). **(b)** When assuming the simple regression derived from panel (a) between precipitation and temperatures, reanalysed coastal precipitation is mostly underestimated (blue areas), while precipitation around Ross Ice Shelf is largely overestimated (red areas).

## 6   Discussion and conclusion

The Clausius-Clapeyron theory suggests a clear relationship between changes in temperature and in the moisture-holding capacity of the air, which can potentially be translated into a relationship between changes in temperature and precipitation. Our study amends the existing literature by analysing the regional and continent-wide scaling factors obtained from the latest available model data from regional model RACMO2.3 and the CMIP6 model ensemble.

Overall, we find that the suite of formerly applied methods to establish the sensitivity of potential precipitation changes in Antarctic for a given amount of warming yield different results. Especially when analysing high-end scenarios with strong changes in annual air temperatures, multi-model mean values can be skewed if the sensitivity factors are calculated through relative changes to a fixed reference period. When using a logarithmic approach for the regression analysis, we generally obtain more robust results, because the Clausius-Clapeyron relationship is logarithmic by nature.

Across all considered SSP scenarios for the period 1850-2100, local scaling factors obtained through grid-point wise regression can exceed 10 % $K^{-1}$, while continent-wide scaling factors from annual mean temperatures and precipitation only yield approximately 5.5 % $K^{-1}$ for all scenarios. This value lies substantially below the theoretical value of 9.45 % $K^{-1}$ obtained for the continent-wide mean annual air temperature of T = 239.55 K for the 20th century reference period. This discrepancy highlights the necessity to use spatially resolved sensitivity factors when scaling local precipitation patterns into the future.

While the change in precipitation in the interior of the Antarctic continent follows the theory quite closely, the scaling factors near the coast can be substantially lower. This can be explained by three particular reasons: First, the presence of a pronounced coastal wind regime can substantially affect local precipitation (Grazioli et al., 2017; Lemonnier et al., 2019).



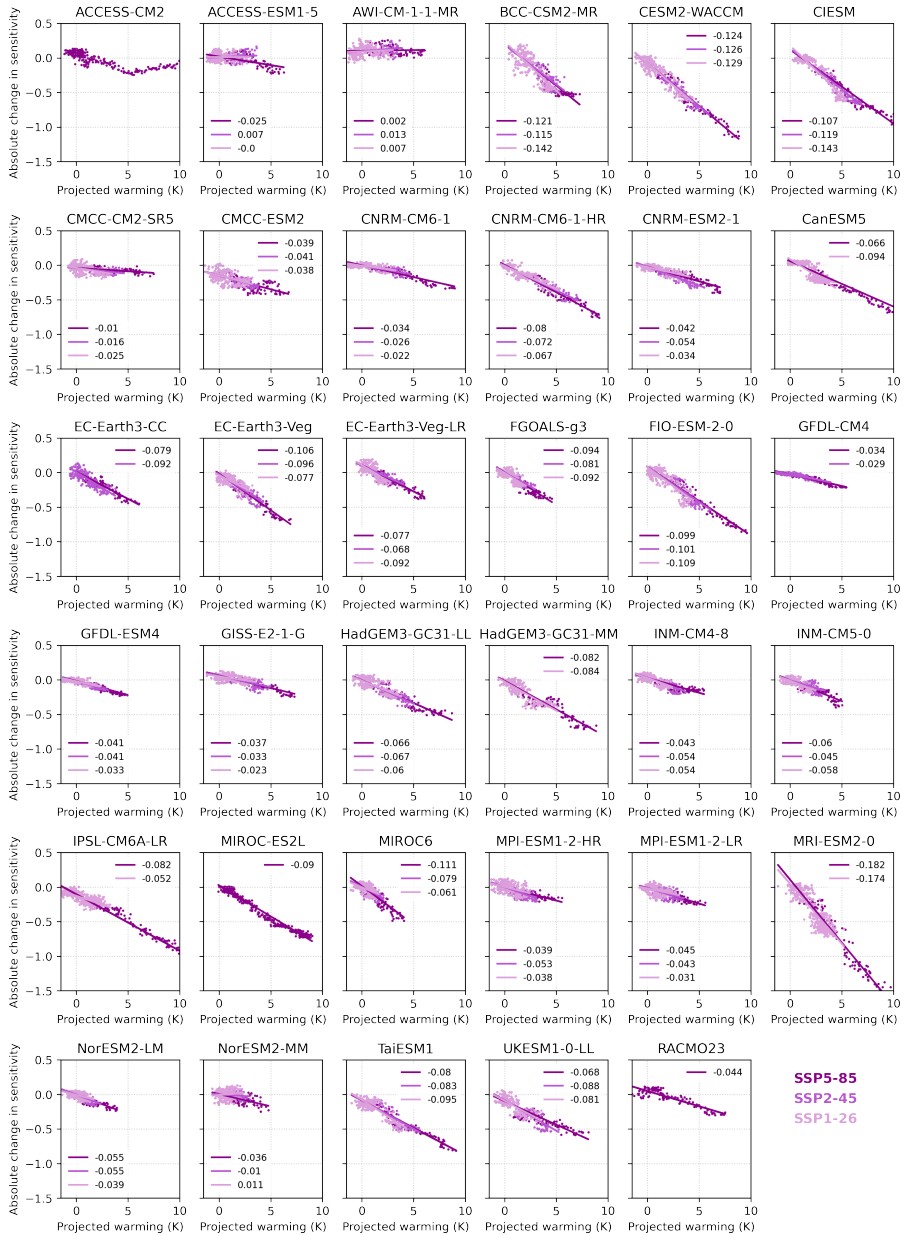

**Figure 7. Decrease in sensitivity factor with warming.** In most models we find for rising temperatures a strong decrease in the sensitivity of Antarctic precipitation to local air temperatures. Results are based on the entire time-series available for each of the individual models.

Second, one of our assumptions is that the available 2 m air temperature data can be used as a proxy for the lower troposphere where the moisture resides. This follows previous studies e.g. Palerme et al. (2017). However, especially the phenomenon of





the near-surface temperature inversion in Antarctica which can amount to 25°C difference between the surface temperature and the lower troposphere in winter (Connolley, 1996), is conflicting with this approach.

A third explanation for the lower sensitivity factors found here could be evaporation constraints, as suggested for instance by Li et al. (2013). Analysing CMIP5 model data, they find that precipitation increases with temperature globally only between 1.5 and 3 % K$^{-1}$. They conclude that one must take into account the energetic constraints on evaporation (approx. 1% – 4

% K$^{-1}$ in the range of 0°C–30°C) when analysing the precipitation scaling globally. We find however that our results do not differ much when analysing net precipitation (precipitation minus evaporation) versus precipitation as done here.

Following Eq.1 we could see a slight decrease in sensitivity factors across the ice sheet depending on the chosen warming scenario in Section 4. This is also confirmed in the spatial regression analysis (see Section 5). This is consistent with the theory of Clausius-Clapeyron, as in colder conditions, for instance in large parts of East Antarctica, the increase of the moisture

holding capacity with warming should be higher when using local conditions.

For the forcing of ice-sheet models, which typically rely on a fixed parameterization with a single sensitivity factor for all temperature ranges, we therefore suggest to introduce temperature-dependent scaling factors, especially for high-end sea-level rise simulations.

Whether – and on which timescales – increased snowfall can offset dynamical ice loss from the Antarctic Ice Sheet in the

future remains very uncertain. For such analysis, one must in particular consider the feedback that snowfall has on the general ice dynamics, since it is known that increased snowfall at the ice-sheet margins enhances the ice flow and thus the ice discharge across the grounding line (Winkelmann et al., 2012). Garbe et al. (2020), using exponentially scaled precipitation, show that despite an increase in surface mass balance, large parts of the Antarctic Ice Sheet could disintegrate on the long-term, with a first critical warming threshold at around 2°C, where the West Antarctic Ice Sheet might become unstable. This means that ice

losses, further accelerated by the marine ice sheet instability (see e.g. Robel et al., 2019), cannot be compensated by additional snowfall as previously assumed.

The assumption that increased snowfall directly translates into an increase in surface mass balance in the future can be further contested by studies investigating the non-linear growth in melt and runoff under warming (Gilbert and Kittel, 2021). Accumulation processes are complex and with increasing melt of snow and of the subsequent firn layer, increased precipitation

hence does not necessarily lead to a mass gain in all parts of Antarctica. Given the present-day temperature conditions, most precipitation falls as snow in Antarctica. With ongoing warming however, rainfall will likely increase in amount, frequency and intensity along the coast of Antarctica over the next 80 years (Vignon et al., 2021). If more precipitation falls as liquid rain, the remaining water on the ice-sheet surface may amplify ongoing surface melt processes through the reduction of surface albedo, latent heat release or hydro-fracturing (Kopp et al., 2017).

For future projections, it will remain important to approximate precipitation increases through temperature-scaling approaches, as coupled simulations with regional climate models remain computationally expensive, especially on multi-centennial timescales. Our results show that these scaling approaches can in principle capture the overall changes of precipitation in a warming world sufficiently well - however, when using a precipitation-scaling approach in ice-sheet modelling studies, the scaling parameter needs to be chosen according to the given application, and its choice should potentially reflect the more complex





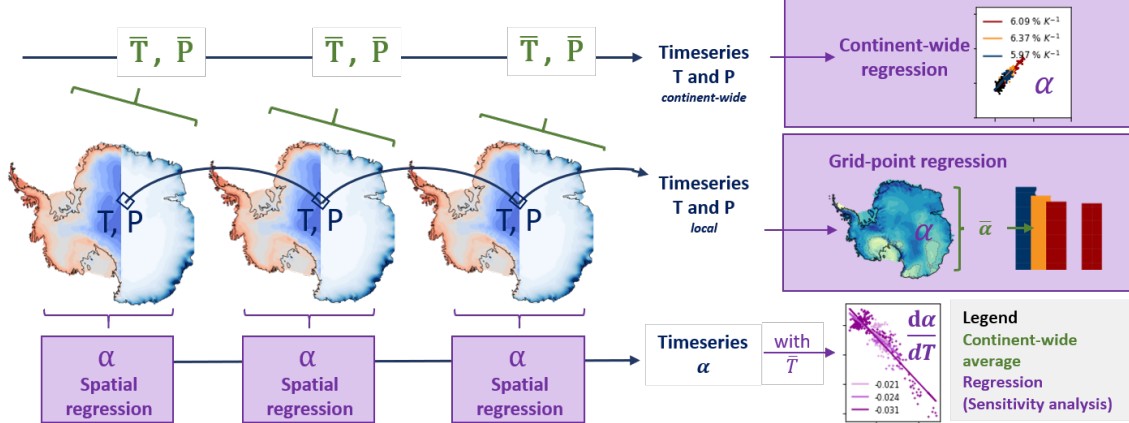

**Figure A1.** Overview of different sensitivity factors estimated in this study.

temperature-dependency outlined here. In particular, our results suggest that Antarctic mass balance projections with uniform estimates of the scaling factor might overestimate the compensating effect of additional snowfall under future warming.

*Data availability.* The CMIP6 data used for this study are freely available from the Earth System Grid Federation (ESGF). We further want to thank the respective authors for providing the RACMO2.3 data that are available upon request.

**Appendix A: Additional figures to the sensitivity analysis**





**Figure A2.** Mean annual air temperature for present-day conditions (1981-2000) from the CMIP6 model ensemble.


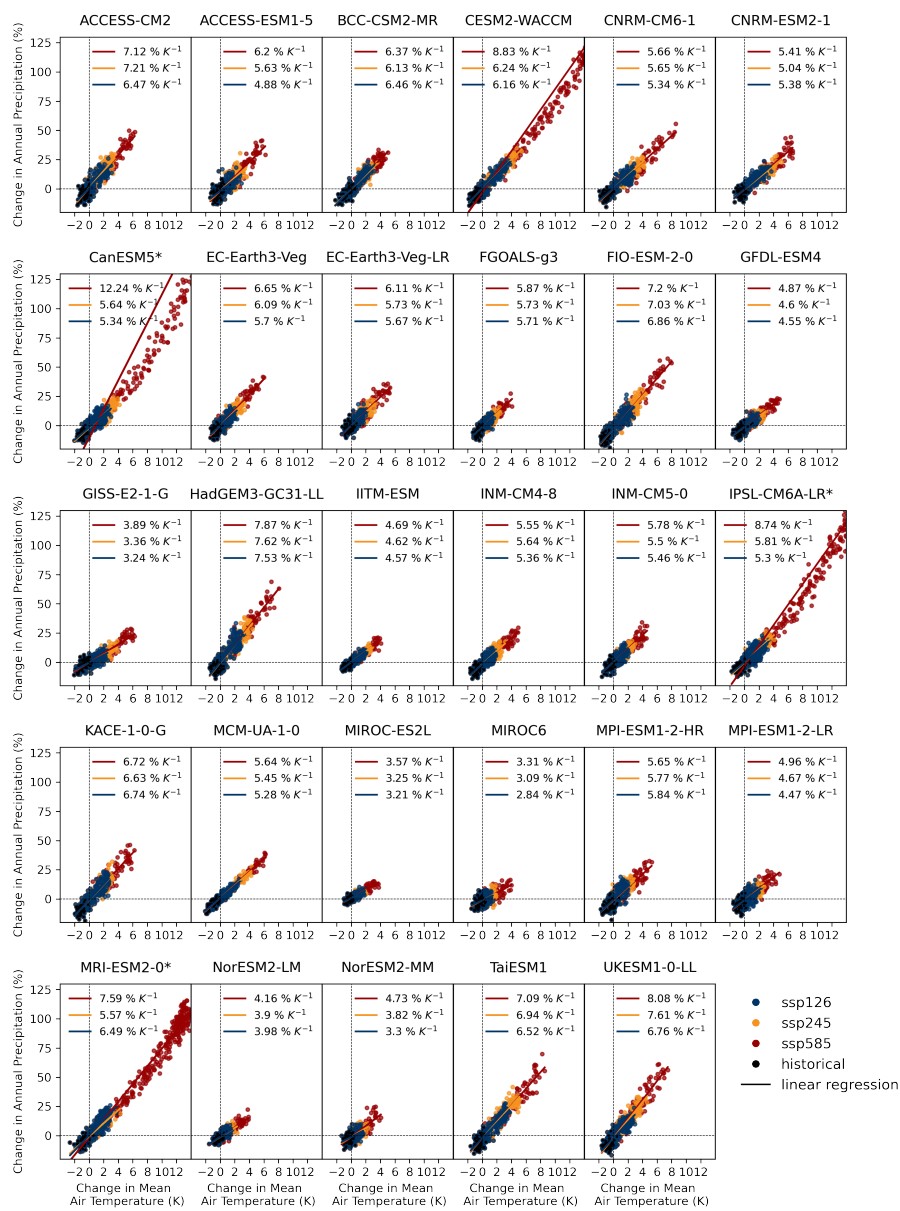

**Figure A3.** Models indicated with * simulate Antarctic climate change up to the year 2300. For the regression analysis the full available time series were used.



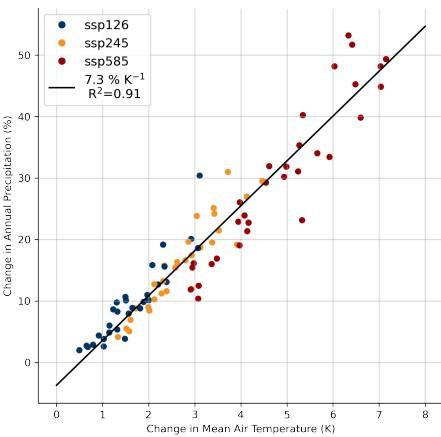

**Figure A4.** Each dot represents an individual model result on how much Antarctic precipitation and temperature has changed between the end of the 20th-century (1981-2000) and the end of the 21st-century (2081-2100).



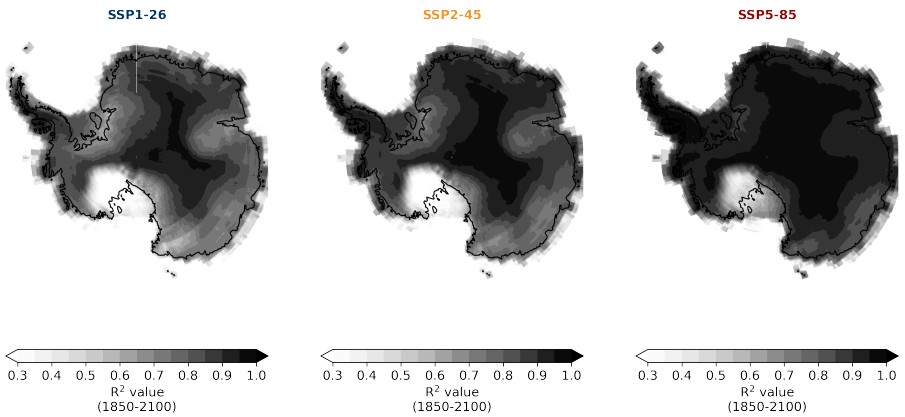

**Figure A5.** $R^2$-values of performed grid-point-wise regression using the CMIP6 ensemble mean of the different SSP-scenarios SSP1-26, SSP2-45 and SSP5-85 (1950-2100).



*Author contributions.* All authors designed the study. LN analysed the data and produced the figures. LN drafted the manuscript with strong support by RW.

*Competing interests.* The authors declare that they have no conflict of interest.

*Acknowledgements.* LN was supported by the Max Planck Institute for Meteorology in Hamburg and stipends by the Potsdam Graduate School and the Studienstiftung des Deutschen Volkes. LN and RW gratefully acknowledge support by the European Union's Horizon 335 2020 research and innovation programme under Grant Agreement No. 820575 (TiPACCs). RW further acknowledges support by the European Union's Horizon 2020 under Grant Agreement No. 869304 (PROTECT), by Deutsche Forschungsgemeinschaft (DFG) through grants WI4556/3-1 and WI4556/5-1 and by the PalMod project (FKZ: 01LP1925D), supported by the German Federal Ministry of Education and Research (BMBF) as a Research for Sustainability initiative (FONA). We carried out our analyses on Mistral, the supercomputer of the German Climate Computing Center (Deutsches Klimarechenzentrum, DKRZ) and its successor Levante.





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



**Table A1. Model-specific results**. We perform a sensitivity analysis with the evolution of log-scaled mean annual precipitation and mean annual air temperature (both continent-wide estimates) over the time period 1850-2100 using different SSP-scenarios.

| Model | hist + SSP1-26 (% K$^{-1}$) | R$^2$ | hist + SSP2-45 (% K$^{-1}$) | R$^2$ | hist + SSP5-85 (% K$^{-1}$) | R$^2$ |
|---|---|---|---|---|---|---|
| ACCESS-CM2 | 5.97 | 0.71 | 6.37 | 0.8 | 6.09 | 0.89 |
| ACCESS-ESM1-5 | 4.39 | 0.5 | 5.08 | 0.66 | 5.14 | 0.85 |
| BCC-CSM2-MR | 6.26 | 0.82 | 5.92 | 0.8 | 5.89 | 0.89 |
| CESM2-WACCM | 5.85 | 0.85 | 5.77 | 0.89 | 5.64 | 0.94 |
| CIESM | 6.17 | 0.84 | 5.86 | 0.88 | 5.55 | 0.93 |
| CMCC-CM2-SR5 | 7.01 | 0.71 | 6.85 | 0.8 | 6.68 | 0.88 |
| CMCC-ESM2 | 6.85 | 0.75 | 6.73 | 0.81 | 6.56 | 0.9 |
| CNRM-CM6-1 | 5.24 | 0.77 | 5.39 | 0.86 | 5.05 | 0.92 |
| CNRM-CM6-1-HR | 6.7 | 0.87 | 6.38 | 0.91 | 5.97 | 0.94 |
| CNRM-ESM2-1 | 5.15 | 0.74 | 4.74 | 0.82 | 4.84 | 0.88 |
| CanESM5 | 5.47 | 0.75 | 5.5 | 0.83 | 5.72 | 0.94 |
| EC-Earth3-CC | n/a | n/a | 5.64 | 0.68 | 5.95 | 0.81 |
| EC-Earth3-Veg | 5.61 | 0.62 | 5.84 | 0.76 | 6.01 | 0.88 |
| EC-Earth3-Veg-LR | 5.52 | 0.56 | 5.45 | 0.7 | 5.52 | 0.82 |
| FGOALS-g3 | 5.76 | 0.63 | 5.69 | 0.73 | 5.64 | 0.84 |
| FIO-ESM-2-0 | 6.9 | 0.8 | 6.8 | 0.87 | 6.43 | 0.93 |
| GFDL-CM4 | n/a | n/a | 5.42 | 0.75 | 5.29 | 0.85 |
| GFDL-ESM4 | 4.74 | 0.56 | 4.73 | 0.65 | 4.84 | 0.8 |
| GISS-E2-1-G | 3.72 | 0.62 | 3.68 | 0.74 | 3.9 | 0.87 |
| HadGEM3-GC31-LL | 7.14 | 0.76 | 7.01 | 0.85 | 6.72 | 0.91 |
| HadGEM3-GC31-MM | 7.33 | 0.83 | n/a | n/a | 6.71 | 0.92 |
| INM-CM4-8 | 5.43 | 0.67 | 5.58 | 0.77 | 5.28 | 0.88 |
| INM-CM5-0 | 5.38 | 0.64 | 5.32 | 0.76 | 5.36 | 0.85 |
| IPSL-CM6A-LR | 5.07 | 0.67 | 5.5 | 0.81 | 5.31 | 0.9 |
| MIROC-ES2L | 3.22 | 0.52 | 3.18 | 0.6 | 3.4 | 0.78 |
| MIROC6 | 2.78 | 0.2 | 2.95 | 0.31 | 3.29 | 0.54 |
| MPI-ESM1-2-HR | 5.78 | 0.55 | 5.63 | 0.68 | 5.29 | 0.78 |
| MPI-ESM1-2-LR | 4.48 | 0.4 | 4.62 | 0.56 | 4.75 | 0.73 |
| MRI-ESM2-0 | 5.35 | 0.72 | 5.39 | 0.79 | 5.42 | 0.89 |
| NorESM2-LM | 4.22 | 0.44 | 4.1 | 0.56 | 4.19 | 0.73 |
| NorESM2-MM | 3.36 | 0.26 | 3.85 | 0.39 | 4.59 | 0.64 |
| TaiESM1 | 6.41 | 0.8 | 6.51 | 0.85 | 6.16 | 0.92 |
| UKESM1-0-LL | 6.73 | 0.73 | 7.23 | 0.84 | 7.08 | 0.9 |
| CMIP6 model mean | 5.48 | 0.65 | 5.46 | 0.74 | 5.46 | 0.85 |
| RACMO2.3 | n/a | n/a | n/a | n/a | 6.37 | 0.92 |