# Peer review of "Revisiting temperature sensitivity: How does Antarctic precipitation change with temperature?"

_The Cryosphere, 2022_

## Author Comment (AC1)

Dear Alexander Robinson, dear reviewers,

We thank the reviewers for their helpful comments and for finding time to examine our manuscript. We are glad to respond to them and change our article accordingly. Our responses are given in blue and italics compared to the comments which are given in black without italic font.

Major changes in the revised manuscript include the following:

- We have restructured the part about the spatial regression and integrated parts of it into the Discussion section.
- We have clarified the language around which regression analysis has been performed.
- We added more discussion around the near-surface inversion and condensation temperature.
- We have removed Figure A1.

On behalf of the authors, and best regards,

**Lena Nicola**

**Response to Referee comment on "Revisiting temperature sensitivity: How does Antarctic precipitation change with temperature?" made by Anonymous Referee #1**

NNW23 (Nicola, Notz and Winkelmann 2023) present new climate model-based estimates of the Antarctic precipitation response to changing temperature. They show there are large spatial differences in the local sensitivity to temperature, with values being greater in the interior and smaller at the Siple coast and Wilkes Land.

The paper provides a very valuable addition to the existing literature on this topic by providing much more spatial detail that is important in projecting the Antarctic contribution to future sea level rise. The paper is clearly suitable for The Cryosphere. I have some comments that I would ask the authors to consider before it can be published.

**Comments**

The estimates given are meant for ice sheet modelers. Still, it would have been nice to provide a little more insight into the meteorology and drivers of these changes, so the reader gets a better sense of what is driving the spatial pattern and how robust it is likely to be in the future. In particular, it would be good to have some discussion about the relative contributions of storm systems vs. saturation vapor pressure in delivering moisture to Antarctica. Only the latter is expected to follow a clean Clausius-Clapeyron relationship to temperature, whereas the former is dependent on the large-scale dynamics of the atmosphere, such as the Southern Annular Mode. Would it be possible to add some discussion (or even analyses?) along these lines?

This is a very interesting point - we have extended the discussion of the relative importance of dynamic drivers to snowfall changes in the revised manuscript. Doing a full analysis on the different contributors, especially storm systems, is beyond the scope of our intended study.

One important aspect of interior Antarctic climate is the near-surface inversion, as the authors briefly acknowledge on lines 288-291. This inversion deserves much more attention that it receives. There are three reasons why this may be important. First, as the authors note, the condensation temperature is much higher than the surface temperature, which means the authors uses temperatures that are too cold in assessing the Clausius Clapeyron relationship. Second, it is well known that the changes in condensation temperature are smaller than those of the surface temperature by a factor of around 0.66 (Connolley, 1996; Jouzel & Merlivat, 1984). This is important when comparing surface tem perature regressions to Clausius-Clapeyron results as the latter would need to be divided by 0.66 to be comparable. Third, the intensification of the inversion with elevation comes from the strong radiative cooling (for this reason the surface lapse rate is super adiabatic). However, CO2 makes the atmosphere less transparent weakening the radiative cooling and changing the relationship between surface and condensation temperature. This complicates the interpretation going forward, and may contribute to the trends observed for high-CO2 scenarios. The authors need to present a more thorough discussion of the inversion layer.

We extended our discussion of the inversion layer within the revised manuscript:

- 1. In the introduction we introduced how condensation temperature can be estimated from surface temperatures, as discussed in the literature. Unfortunately, a full analysis of the inversion layer, of the relationship between condensation temperature and surface temperature, and of their influence on the sensitivity factors is beyond the scope of this study. We therefore now provide preliminary estimates of their impact on our results, and highlight the necessity of future analyses on this point.
- 2. In the discussion section we now discuss in detail the additional points made by the reviewer, such as the CO2 relationship.

One of the best papers written on this topic is (Fortuin & Oerlemans, 1990). They have a very clear discussion of the spatial patterns, the sections of the ice sheet where saturation vapor pressure dominates, and the role of the inversion. They authors should cite this work – and more importantly study it carefully as I think it would really help them in addressing some of my earlier comments.

Interestingly, that paper already found that there is broadly a greater sensitivity in interior Antarctica than on the coasts (Table V), which is exactly the conclusion of NNW23!! At the very least the study should be included in Figure 1.

We thank the reviewer for pointing us to Fortuin & Oerlemans, 1990 which indeed fits very well to our study. We added a respective entry in Figure 1 and discuss it in comparison to our findings in the text.

Throughout the paper it would be helpful if the language around "regression" was clarified and made more consistent. Since all the analyses performed are essentially regression analyses, it would be helpful if the language were more specific. For example state clearly each time whether you are performing spatial or temporal regression, and whether you are performing regression on a linear or logarithmic scale.

**We have clarified the use of the word "regression" throughout the revised manuscript.**

On lines 119-121 the authors argue that the regression should be performed against the log of precipitation, and I could not agree more (this has always bothered me about earlier papers on this topic). However, it seems that the authors do perform linear regression after all (I think they call it "Regression using anomalies"). This is unnecessarily confusing to me. Please consider removing the linear regression from the paper – I think it is enough to just state that this explains your difference from Frieler et al. without going into too much detail.

**Agreed - we have revised this section and shortened the comparison.**

The motivation behind section 5 was confusing to me. It seems like this is the only section where a spatial regression was used (correct?). Maybe it should be called "spatial regression" instead. It is clear that spatial regression is the least accurate of the methods used, so why is this method selected to investigate the decrease in sensitivity with future warming? To me it would make a lot more sense to have a separate section on "spatial regression", and a section on the "decrease in sensitivity with future warming" using temporal regression instead – the thinking being that temporal regression is more accurate than spatial regression.

We agree that the spatial regression is the least accurate method in this case. We have therefore restructured the text, so that we now have a clear focus on the temporal regression analysis, and only include the spatial regression in the discussion section. In a related point, to do a spatial regression well, one should consider additional factors such as surface slope that drives the orographic uplift. Fortuin and Oerlemans (1990) could provide a template for more meaningful spatial regression.

As the reviewer rightly points out that the spatial regression is the least accurate method, we now decided to minimise its analyses in the manuscript.

On the difference between the continent-wide and grid-point regression (line 235 onward), it seems to me that this just reflects the fact that in the continent-wide sensitivity is effectively weighted by accumulation rate (continent-wide total precip is dominated by high-acc regions), whereas the grid-point sensitivity is weighted by surface area (dominated by inland low-acc regions). Since the inland low-acc regions tend to have higher sensitivity, this difference is exactly what I would have expected. Please comment.

**In both cases, the averages are weighted by surface area. We have clarified this in the manuscript.**

As a last comment, the authors compare many different maps of sensitivity and discuss their differences. Practically, it would be of great value if the authors could come up with a recommended map of sensitivity for future ice modeling studies to use, or a set of recommended procedures for ice modelers and other interested parties. That way modelers could download this recommended set of sensitivities and apply it to their work without having to redo this study or having to make decisions about what set is best.

We are currently preparing the data product to be uploaded on Zenodo and added a recommendation in the discussion. Generally, the most important recommendation would be to adapt the scaling factors according to the forcing scenario (lower sensitivities for high-end warming scenarios).

**Line-by-line comments:**

Line 11: I think instead of 7% this should be "7 percentage points" *Corrected.*

Line 34-35: Please use one unit for snowfall consistently through the paper and specify it here. In this one sentence you use 171 mm and 186 mm w.e. – are these different or the same?.

They are the same. We clarified the unit in the revised manuscript.

Line 45: this would be a good place to introduce Fortuin and Oerlemans 1990 *We now refer to it in the revised manuscript.*

Line 53: what is e\_s? Specify In the manuscript we specified that e\_s is the saturation water vapour pressure.

Line 55: why do you use -13C for a global approximation? Mean global temp is +14

We have revised this section in the manuscript.

Line 58: this would be a good place to specify that it's the condensation temperature that matters and not the surface – what value would you get from Clausius-Clapeyron for the TC? (using the relationship between TS and TC from s?

We added such discussion.

Fig. 1a: could you plot the precip on a log scale instead? Fig. 1b: add Fortuin and Oerlemans (1990) *Both comments are addressed in the new figure 1.*

Around line 98: When it comes to ice cores, an important paper is also (van Ommen et al., 2004). This paper suggests enormous contributions from storms rather than saturation vapor pressure, hinting at the complexities in the T-A relationship.

We include this paper in our discussion section.

Line 150 onward: Do you spell out the acronyms somewhere – in a figure caption for example?

We added "shared-socioeconomic pathway (SSP) - scenarios" for clarification. What the individual scenarios such as SSP1-26 in detail stand for, we would like to refer to Riahi et al., 2017.

Line 152: r1i1f1p1 means nothing to most readers – only to the experts. Clarify or remove

Removed.

Line 192: this high number of 13.31 comes from a linear fit correct? I would remove this whole section, as well as Fig. 3a as it mostly confuses the reader. Linear regression is not listed in your list of 3 regression methods (methods section), which is why I got confused. *We removed Figure 3a.*

Line 235-236: I think this difference is because the continent-wide estimates are biased toward the high accumulation regions. *See above comment. All averages are weighted by area.*

Fig 4: why are there no estimates over the ice shelves? Some CMIP models mask the ice shelves in their output - for consistency, we here only include the area which results from the intersection of all native masks (i.e., where all CMIP models have ice). It is true that this means in some cases high accumulation areas are excluded.

Fig 4: Accumulation on the Siple coast and Ross ice shelf have been suggested to have a strong ENSO imprint on them – could that be the reason for the low sensitivity here?

We have taken up this point in Chapter 4.

Section 5: As suggested before, this section is more about spatial regression. Why use this imperfect tool to investigate changing sensitivity? We have restructured our manuscript, so that we briefly discuss the spatial discussion when evaluating our findings in Section 5 of the revised manuscript.

Line 271: Clausius-Clapeyron equation (not theory) *Corrected.*

Line 297: Could changes to the inversion structure play a role here also? Clausius Clapeyron should be evaluated at the condensation temp, not surface temp, and the relationship between these two is more complex. *We added more discussion to this in the revised manuscript.*  Line 309: what is the threshold of 2 degree refer to? 2 degrees of global warming? SST warming around Antarctica? Antarctic surface warming? *It refers to 2°C of global warming above pre-industrial levels. We clarified it in the revised manuscript.*

Data availability: I think the authors should make the spatial maps of the temperature sensitivity from the grid-based evaluations available. That will be a valuable resource for future studies, and modeling efforts.

*The spatial maps of the temperature sensitivity from Figure 4 will be made available on Zenodo.*

Fig A1: I really don't understand this figure, or what it is trying to communicate. Can you please clarify or rework the figure? Why are there three identical pictures or Antarctica on the left with three times the same text? Figure A1 was intended to help illustrate the different methods. As it is apparently very unclear and confusing, we will hence remove the figure entirely and hope that our methodology needs no further illustration.

**References:**

Connolley, W. (1996). The Antarctic temperature inversion. International Journal of Climatology: A Journal of the Royal Meteorological Society, 16(12), 1333-1342.

Fortuin, J., & Oerlemans, J. (1990). Parameterization of the annual surface temperature and mass balance of Antarctica. Ann. Glaciol, 14, 78-84.

Jouzel, J., & Merlivat, L. (1984). Deuterium and oxygen 18 in precipitation: Modeling of the isotopic effects during snow formation. Journal of Geophysical Research: Atmospheres, 89(D7), 11749-11757. https://doi.org/10.1029/JD089iD07p11749

van Ommen, T. D., Morgan, V., & Curran, M. A. J. (2004). Deglacial and Holocene changes in accumulation at Law Dome, East Antarctica. Annals of Glaciology, 39(1), 359-365.

http://www.ingentaconnect.com/content/igsoc/agl/2004/00000039/00000001/a rt00055

http://dx.doi.org/10.3189/17275640478181422

**Response to Referee comment on "Revisiting temperature sensitivity: How does Antarctic precipitation change with temperature?" made by Anonymous Referee #2**

From a technial point of view I don't have anything to add beyond the very nice Review 1. I similarly did not understand the value of Figure A1. *We removed Figure A1 from the manuscript.*

Philosophically however, the precipitation scaling approach is driven by the needs of ice sheet modelers for very simple mass input on very long time scales. Reading lines 312-319 about precipitation complexities that are emerging as a result of recent research says to me that such simple scaling approaches are liable to yield wrong answers on at least the 100 year time scale.

As one example, the recent large variability in Antarctic sea ice variability since 2016 (https://doi.org/10.1038/s41561-021-00768-3) underlines the multidecadal variability impact on Antarctic climate via the IPO and AMO, etc. Where the sea ice edge resides impacts the continental temperatures and precipitations. Time is rapidly approaching when ice sheets should be treated as fully coupled components of global earth system models even for multimillenial integrations.

We have taken up the point in the revised manuscript, adding that this underlines the calls for further improving global earth system models that treat ice sheets as fully coupled components especially for multi-millennial integrations.

---

## Author Response (AR2)

*Dear Alexander Robinson,*

*Thank you again for finding the time to examine our manuscript. We have taken up all your recommended changes and adjusted our article accordingly. Our responses are given in blue and italics compared to the comments which are given* in black without italic font.

*Best regards,*

*Lena Nicola*

To the authors,

You have done a thorough job responding to the reviewers' comments, which were largely positive. I believe the manuscript is near ready for publication, following some minor additional changes, listed below.

Best regards,
Alex

Minor comments:

L2: "discharge, calving and melting" <= I think here you should either refer to "discharge and melting" or "calving and melting". In this context, discharge and calving are somewhat redundant.
*Corrected to "ice discharge and melting".*

L18: risen => raised
*Corrected.*

L37: is exceeding => exceeds
*Corrected.*

L49-50: condensates => condenses
*Corrected.*

L58: such calculation => such a calculation
*Corrected.*

L57: "$T_g$ = -33.6∘C" <= I would just put Kelvin here directly, since this is the unit used throughout the manuscript, and even later in the paragraph. If you want to put degrees Celcius for reference, put it in the parentheses, instead of the other way around.
*Replaced as recommended, but kept the degrees Celcius as reference.*

L87: scenarios (SSP5-85). => scenarios (SSP5-85), respectively.
*Corrected.*

L87: For these => For each of these
*Corrected.*

L88: 28.8 ± 12.6 %, respectively. => 28.8 ± 12.6 %.
*Corrected.*

L103: catabatic => katabatic
*Corrected.*

L108: Such increase => Such an increase
*Corrected.*

L113: among others => among other things
*Corrected.*

L136: is changing => changes
*Corrected.*

L173: from Earth System Grid Federation => from the Earth System Grid Federation
*Corrected.*

L209: in parts => in part
*Corrected.*

L213: 20th <= superscript "th". Also check throughout the manuscript for 20th and 21st, etc.
*Corrected throughout the text (see revised manuscript).*

L240: ice dynamical => ice-dynamics based
*Corrected.*

L311: This gets evident => This becomes evident
*Corrected.*

L324-325: are affecting => affect
*Corrected.*

L327: should therefore => should
*Corrected.*

L330: the model => the models
*Corrected.*

L360: dynamical => dynamic
*Corrected.*

L361: For such analysis => For such an analysis

*Corrected.*